# Trypsin may be associated with duodenal eosinophils through the expression of PAR2 in early chronic pancreatitis and functional dyspepsia with pancreatic enzyme abnormalities

**Shuhei Agawa[1], Seiji Futagami[1]\*, Hiroshi Yamawaki[1], Rina Tsushima[1], Kazutoshi Higuchi[1], Mayu Habiro[1], Rie Kawawa[1], Yasuhiro Kodaka[1], Nobue Ueki[1], Yoshiyuki Watanabe[2], Katya Gudis[1], Rhuji Ohashi[3], Katsuhiko Iwakiri[1]**

1 Department of Gastroenterology, Nippon Medical School, Tokyo, Japan, 2 Department of Internal Medicine, Kawasaki Rinko General Hospital, Kawasaki, Japan, 3 Department of Diagnostic Pathology, Nippon Medical School, Tokyo, Japan

\* seiji.futagami@gmail.com

## Abstract

### Background

Early chronic pancreatitis (ECP) has been reported to advance into chronic pancreatitis, it may be critical to differentiate the pathophysiology of ECP and functional dyspepsia (FD) in patients with pancreatic enzyme abnormalities (FD-P). This study aimed to clarify differences in the pathophysiology of ECP and FD-P and to determine whether duodenal inflammatory responses in the two diseases were associated with protease-activated receptor (PAR) 2, as the trypsin receptor.

### Methods

Eighty patients who presented with FD-P and ECP were enrolled. In duodenal specimens, PAR2 mRNA levels were determined using real-time PCR. Using immunostaining, CD68-, GLP-1-, PRG2-, and CCR2-positive cells, tight junction proteins, and PAR 2 were evaluated.

### Results

There were no significant differences in clinical symptoms and gastric motility between ECP and FD-P patients. The CD68-positive cells infiltrations and occludin expression levels in the duodenal mucosa of patients with FD-P were significantly ($p<0.001$ and $p = 0.048$, respectively) lower than those in patients with ECP. Although serum trypsin levels in ECP and FD-P patents were significantly ($p<0.05$ and $p<0.001$, respectively) associated with duodenal eosinophils counts, elevated trypsin levels were not significantly associated with degranulated eosinophils, occludin, claudin-1 and ZO-1 expression levels in the duodenum of either group. PAR2 mRNA levels were increased in the duodenum of patients with ECP

**Data Availability Statement:** All relevant data are within the manuscript and its Supporting Information files.

**Funding:** The Ministry of Education, Culture, and Science and the Ministry of Health, Japan (16K09294) funded this study. Prof. Futagami acquired the grant. The funders had no role in study design, data collection and analysis, decision to publish, or preparation of the manuscript.

**Competing interests:** The authors have declared that no competing interest exist. I have read the journal's policy and the authors of this manuscript have the following competing interests.

and FD-P. PAR2 was localized in the epithelial cells of the duodenal mucosa and the surface of degranulated eosinophils in ECP and FD-P patients.

## Conclusions

Elevated trypsin levels might be partly associated with duodenal inflammatory responses through PAR2-related degranulated eosinophils and the reduction of occludin in patients with ECP and FD-P.

## Introduction

The Rome III classification defines functional dyspepsia (FD) patients based on their bothersome abdominal symptoms despite the absence of organic diseases using abdominal ultrasonography or abdominal CT scanning [1,2]. FD can also encompass chronic pancreatitis who are complicated with abdominal fullness, epigastric pain, and early satiety [3]. Andersen et al. and Smith et al. reported that a portion of the population of FD patients had pancreatic enzyme abnormalities and chronic pancreatitis [4,5]. In 2015, the Japanese Society of Gastroenterology (JSGE) reported new guidelines for FD and demonstrated that among FD patients, 24% had chronic pancreatitis (CP) [4,6]. Sahai et al. have also studied that dyspepsia may be an atypical presentation of pancreatic disease as determined by endoscopic ultrasonography (EUS) [3]. In Japan, to prevent the early phase of CP from advancing into CP, new strategies for addressing CP in its initial stages have been proposed [7]. According to the Japan Pancreatic Association (JPA), four clinical criteria including epigastric pain and the presence of more than two features of EUS are needed for a diagnosis of early chronic pancreatitis (ECP) and ECP has been reported to be a high-risk group for advanced CP and pancreatic cancer [7]. We recently conducted a study comparing the clinical characteristics and duodenal inflammation between patients with ECP and FD with pancreatic enzyme abnormalities (FD-P), and both groups had very similar characteristics [8–10]. Because the appropriate treatment for ECP based on a precise diagnosis can improve refractory FD-like symptoms and endosonographic findings [10,11], it is critical to differentiate the pathophysiology between ECP and FD-P patients to prevent progression to CP and pancreatic cancer.

Several studies have reported that duodenal inflammation was linked to FD patients [12–14]. We have also recently reported that duodenal degranulated eosinophils significantly increased in FD-P patients compared to ECP patients [11]. Larauche et al. have reported that the trypsin receptor, protease-activated receptor 2 (PAR2) which plays a critical role in visceral hypersensitivity in irritable bowel syndrome patients [15]. Kong et al have also reported that trypsin regulates enterocytes through the activation of PAR2 [16]. In addition, activation of PAR2 has been linked to mucosal inflammation [17,18]. In this study, we compared trypsin levels and tight junction proteins, such as occludin, in duodenal inflammatory cells of ECP and FD-P patients.

Thus, since it is critical to treat with ECP and FD-P patients based on the distinct pathophysiology of two groups and to clarify whether pancreatic enzyme abnormalities were associated with duodenal inflammatory responses and refractory symptoms, we tried to analyze whether pancreatic enzyme levels such as trypsin levels affected the expression levels of tight junction proteins or duodenal inflammatory cells infiltrations such as eosinophils through the expression of PAR2 in the duodenum of patients with ECP and FD-P.

## Material and methods

### Patients

This study enrolled 80 consecutive patients from April 2013 to March 2020, who presented with ECP (n = 26), functional dyspepsia with pancreatic enzyme abnormalities (FD-P) (n = 54) after the measurement of pancreatic enzymes, upper GI endoscopy, abdominal ultrasonography and abdominal computed tomography. Asymptomatic patients without abnormal images (n = 12) were enrolled as controls. According to the none of pancreatic enzyme abnormalities, we first diagnosed FD patients and later the others were divided with the endosonographic findings in ECP and FD-P patients (S1 Fig). Patients with FD were determined by the Rome III criteria [1]. Exclusion criteria included patients with severe heart disease, renal or pulmonary failure, liver cirrhosis, severe systemic illness, and a history of malignant disease. *Helicobacter pylori* (*H. pylori*) infection was determined by both the $^{13}$C-urea breath test and by measurement of anti-*H. pylori* antibody. We measured five kinds of pancreatic enzymes in the sera of FD patients. Written informed consent was obtained from all subjects prior to undergoing upper GI endoscopy and abdominal ultrasonography for the evaluation of dyspeptic symptoms. The study protocol was approved by the Ethics Review Committee of Nippon Medical School Hospital (556-2-21).

### Analysis of serum pancreatic enzymes and duodenal trypsin contents

Serum levels of trypsin, PLA2, lipase, p-amylase, and elastase-1, and trypsin levels in the duodenal fluids taken from endosonographys were analyzed using an automated chemistry analyzer (AU 5822 analyzer; Beckman Coulter, Brea, CA, USA).

### Clinical symptoms

The clinical symptoms of FD were estimated based on the Rome III criteria [1]. The diagnoses of PDS and EPS were determined with symptoms occurring for the last three months and the onset of symptoms occurring at least six months prior to diagnosis. In this study, each FD symptom was estimated based on the Rome III classification [1] and evaluated as follows: 0, none; 1, very mild; 2, mild; 3, moderate; 4, severe; and 5, very severe. Clinical symptoms were estimated using the Gastrointestinal Symptom Rating Scale (GSRS) [19]. Dyspeptic symptoms were evaluated using the mean GSRS and the 15 GI symptoms of the GSRS.

### Clinical symptoms for fat intakes

In this study, high-fat meals were defined as those containing more than 16 grams of fat per 100 grams of food according to previous study [20]. Patients consuming high-fat meals evaluated their own clinical symptom scores as follows: 0, no complaints; and 1, presence of any clinical symptoms including gastroesophageal reflux, abdominal pain, dyspepsia, diarrhea and constipation.

### Endosonographic assessment

An Olympus EUS-UCT 260 convex scanning endosonography (Olympus America, Melville, NY, USA) was used to perform EUS under conscious sedation in 80 FD patients with pancreatic enzyme abnormalities (S1 Fig). Endosonographic features were determined as previous study [7]. When there were any differences in opinions among the expert endoscopists, the final diagnosis was determined by consensus following a discussion of each case. Early chronic pancreatitis cases were diagnosed based on the presence of two or more EUS imaging findings, among the seven findings listed above, and the presence of two or more clinical symptoms or

findings, including repeated attacks of epigastric pain, abnormalities in blood/urine pancreatic enzymes, exocrine pancreatic dysfunction and a chronic alcohol intake (80g/day) (S1 Fig).

## RNA extraction and real-time PCR assay for PAR2

Duodenal tissues were first homogenized and the RNA was extracted from duodenal mucosa samples using an RNA extraction kit (QIAGEN, Venlo, Netherlands). RNA was stored at –80 degrees until use. RNA (2 μL) was applied in an RNA-to-cDNA kit (Thermo Fisher Scientific). The reaction mixture was incubated at 37˚C for 60 minutes, and stopped by heating to 95˚C for 5 minutes and then held at 4˚C. After completion, the cDNA was stored at -20˚C. TaqMan Advanced Master Mix (18μl) and the Primer Probe Mixture (1μl) were added to 1 μL of RNA for a total volume of 20 μL. Primers and probes used for real-time PCR were created using the PAR2 TaqMan Gene Expression Assay (Thermo Fisher Scientific, MA, USA). The primers and probes for PAR2 (F2RL1) (accession number: Hs00608346_m1*) and GAPDH (accession number: Hs99999905_m1*) were purchased from Thermo Fisher Scientific.

## Evaluation of duodenal inflammatory cells and tight junction proteins

Histological duodenitis was estimated as mild, moderate, or severe by H&E staining according to the criteria of previous studies [21,22]. Mild duodenitis was defined as an expansion of the lamina propria by mild inflammatory cell infiltration. Moderate duodenitis was determined by the partial loss of villi and expansion of the lamina propria by moderate inflammatory cell infiltration. Severe duodenitis was estimated by the partial loss of villi and expansion of the lamina propria by severe inflammatory cell infiltration, mainly consisting of plasma cells, macrophages, and lymphocytes. Specimens were estimated by two expert pathologists in a blinded manner. Migrated eosinophils were evaluated using HE staining.

CD68-positive cells, CCR2-positive cells, GLP-1 positive cells or degranulated eosinophils were analyzed by immunostaining using a mouse anti-CD68 antibody (DAKO, Glostrup, Denmark: 1:100), goat anti-CCR2 antibody (Abcam, CA, UK:1:200), rabbit anti-GLP-1 antibody (Abcam, Cambridge, UK; 1:2000) and mouse anti-PRG2 antibody (Gene Tex, CA, USA: 1:50), respectively.

In addition, tight junction proteins including claudin-1, occludin and ZO-1 were estimated by a mouse anti-claudin-1 antibody (Thermo Fisher Scientific, MA, USA: 1:200), rabbit anti-occludin antibody (Abcam, CA, UK: 1:500), rabbit anti-ZO-1 antibody (Abcam, CA, UK: 1:30), and rabbit anti-PAR2 antibody (NOVUS, CO, USA: 1:100), respectively.

Double-labeling immunofluorescence methods and confocal laser scanning microscopy were used to evaluate the coexpression of immunoreactivity for the pair of rabbit anti-EpCAM antibody (Abcam, CA, UK: 1:20) and mouse anti-PAR2 antibody (Santa Cruz, TX, USA: 1:20). In addition, we also determined the expression of immunoreactivity for the pair of mouse anti-PRG2 antibody (Gene, Tex, USA: 1:10) and rabbit anti-PAR2 antibody (NOVUS, CO, USA:1:10). Sections were incubated overnight at 4˚C with a mixture of the two primary antibodies and then with FITC or Texas red-conjugated secondary antibodies [horse anti-mouse IgG (Vector Laboratories, CA, USA) and goat anti-rabbit IgG (Vector Laboratories), for EpCAM and PAR2 or PRG2 and PAR2, respectively] followed by nuclear counterstaining with VECTSHIELD HardSet mounting medium with DAPI (Vector Laboratories).

## Scoring of immunostaining for tight junction proteins in the duodenum

The expression levels of tight junction proteins were assessed by multiplying the scores according to a modified version of a previous study [23]. Briefly, each score between 1 and 3 was given to classify the percentage of positive cells (1, <25%; 2, 25–50%; 3, >50%) and the

intensity of membrane staining (1: mild staining, 2: moderate staining, 3: intense staining), respectively. The expression level of tight junction proteins (from 1 to 9) was defined by multiplying above two scores for positivity and intensity.

## Measurement of gastric emptying

Sodium acetate (water soluble; $^{13}$C-acetate) was used as a tracer for the emptying of liquids (Cambridge Isotope Laboratories; Tewksbury, MA, USA). We used an integrated software solution program to calculate the half-gastric emptying time ($T_{1/2}$) and the lag phase (Tmax; min) based on a previous study [24]. A Tmax value greater than 60 min, (representing the mean Tmax in healthy volunteers plus SD), was defined to identify relative disturbances in gastric emptying according to the diagnostic criteria of the Japan Society of Smooth Muscle Research and our study [13,25].

We determined the area under the curve at 5 minutes ($AUC_5$) and 15 minutes ($AUC_{15}$) values as markers of the early phase of gastric emptying based on previous studies [26,27].

## Sample size

The sample size was determined using the Power and Sample Size Calculation Program (PS), gifted to us from Vanderbilt University. The standard deviation of the expression levels of occludin in ECP patients was approximately 1.8 ($\sigma = 1.8$). Using the above data, and setting $\alpha = 0.05$, $\beta = 0.80$, and the estimated mean of the expression levels of occludin in patients with ECP = 4.67, it was determined that 26ECP patients and 33FD-P patients were estimated to be sufficient to identify clinically relevant differences.

## Statistical analysis

Statistical analysis was almost performed using two-tailed unpaired t-tests. The percentage of pancreatic enzyme abnormalities and endosonographic findings were performed using chi-square ($\chi 2$) test. We examined tight junction proteins and inflammatory cells, and duodenal trypsin levels and serum trypsin levels using correlation coefficients. All analyses were performed using the standard software package (SPSS version 27.0, IBM Corp., Armonk, NY, USA) and p-values <0.05 were considered significant.

## Results

### Clinical characteristics of ECP and FD-P patients

Age, sex, BMI, *H. pylori* positivity, and total GSRS score did not differ significantly between FD-P and ECP patients (Table 1). In addition, there were no significant differences in $HbA_{1C}$ values or proportion of peripheral eosinophils. On the other hand, immunoreactive insulin (IRI) values were significantly (p = 0.045) higher in patients with FD-P than in those with ECP (Table 1). In addition, positivity of abnormal PLA2 levels in FD-P patients was significantly (p = 0.028) higher compared to ECP patients. There were no significant differences in p-amylase, lipase, trypsin, and elastase-1 levels between ECP and FD-P groups (Table 1). Positivity of abnormal trypsin level was highest among all pancreatic enzymes (Table 1).

### Comparison of clinical symptoms between ECP and FD-P patients

To determine whether there were any differences in clinical symptoms between ECP and FD-P patients, we compared bothersome FD symptoms, feeling of hunger, and discomfort for fat intake between ECP and FD patients. There were no differences in epigastric pain,

**Table 1. Characteristics of ECP and FD-P patients.**

| Factor | FD-P (n = 54) | ECP (n = 26) | *P* value |
|---|---|---|---|
| age | 55.3±2.26 | 59.2±2.96 | 0.318 |
| sex(M/F) | 25/29 | 12/13 | 0.890 |
| BMI | 21.62±0.406 | 21.08±0.708 | 0.477 |
| GSRS | 2.539±0.189 | 2.563±0.265 | 0.942 |
| HbA1c | 5.604±0.0489 | 5.78±0.237 | 0.492 |
| IRI | 11.616±1.38 | 8.87±1.802 | 0.045 |
| Eosinophil(%) | 3.30±0.36 | 2.81±1.07 | 0.648 |
| trypsin (%) | 81.48 | 78.26 | 0.152 |
| PLA2 (%) | 51.85 | 47.83 | 0.028 |
| lipase (%) | 33.33 | 45.45 | 0.783 |
| p-amylase (%) | 41.51 | 18.18 | 0.139 |
| elastase-1 (%) | 9.26 | 13.04 | 0.343 |
| *H. pylori* positivity(%) | 3.7 | 3.85 | 0.975 |

ECP: Early chronic pancreatitis, FD-P: Functional dyspepsia with pancreatic enzyme abnormalities.

GSRS: Gastrointestinal Symptom Rating Scale, IRI: Insulin resistance index.

epigastric burning, postprandial fullness, early satiety, abdominal fullness, the feeling of hunger, or discomfort for fat intake between the two groups (Fig 1A).

## Comparison of Gastric Motility Between ECP Patients and FD-P Patients

To clarify whether there were significant differences in gastric emptying between ECP and FD-P patients, we measured Tmax and $T_{1/2}$ values such as gastric emptying, and $AUC_5$ and $AUC_{15}$ values such as the early phase of gastric emptying between ECP and FD-P patients. There were no significant differences in Tmax and $T_{1/2}$ values between ECP and FD-P patients (Fig 1B). In addition, there were no significant differences in $AUC_5$ and $AUC_{15}$ values between ECP and FD-P patients (Fig 1B).

In addition, there were no significant differences in AUC5 and AUC15 values between ECP and FD-P patients. We compared these using a two-tailed, unpaired, t-tests.

## Comparison of endosonographic features between ECP and FD-P patients

We compared seven items of endosonographic features between patients with FD-P and those with ECP. Hyperechoic main pancreatic duct margin (14/26), lobularity with honeycombing (1/26), stranding (23/26), dilated side branches (3/26) and hyperechoic foci without shadowing (13/26) in ECP patients were significantly (p<0.001, p<0.001, p<0.001, p = 0.011 and p<0.001, respectively) higher than those in FD-P patients (Fig 2A). Representative endosonographic features of patients with ECP are shown in Figure (Fig 2B).

## Comparison of tight junction proteins expression in the duodenal mucosa between FD-P and ECP patients

Increased mucosal permeability is associated with changes in tight junction proteins in patients with FD. Thus, we compared the expression levels of occludin, claudin-1, and ZO-1 proteins in the duodenal mucosa of the patients with FD-P and ECP patients. The expression level of occludin in the duodenum of FD-P patients (1.55±0.098) was significantly reduced compared to that in ECP patients (1.85±0.11, p = 0.043) (Fig 3A). However, there were no

A

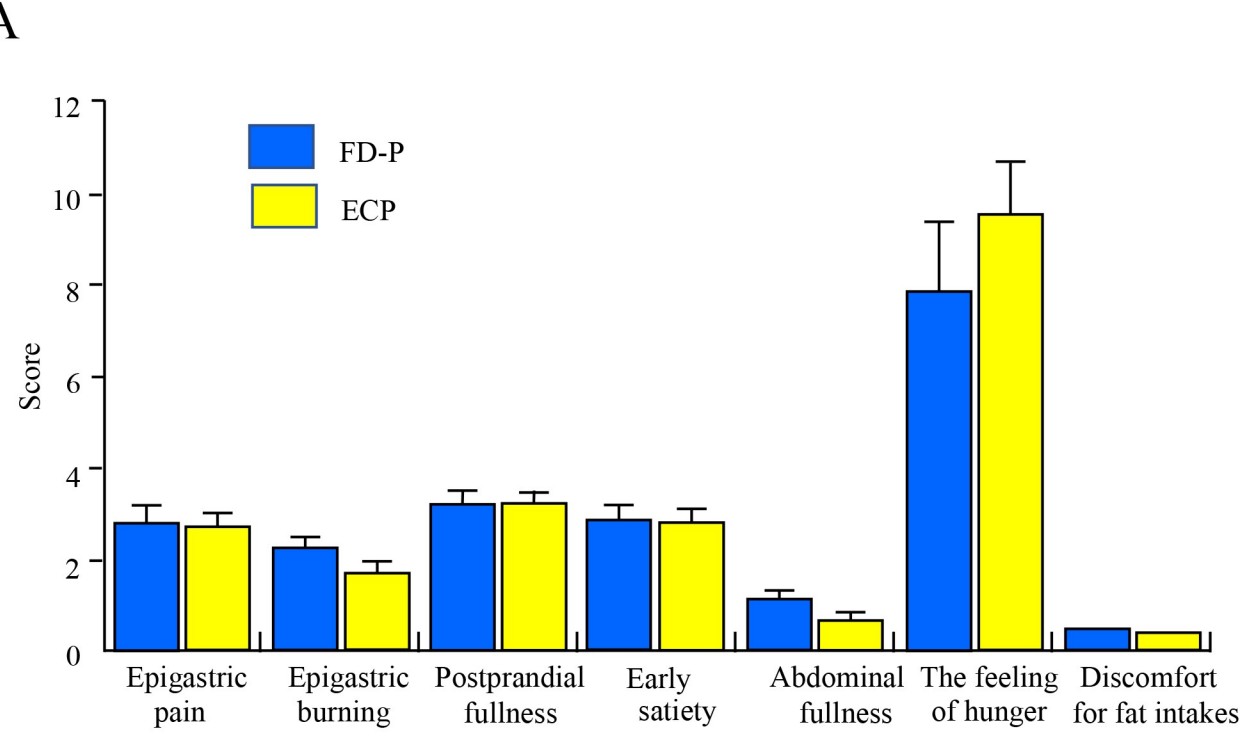

B

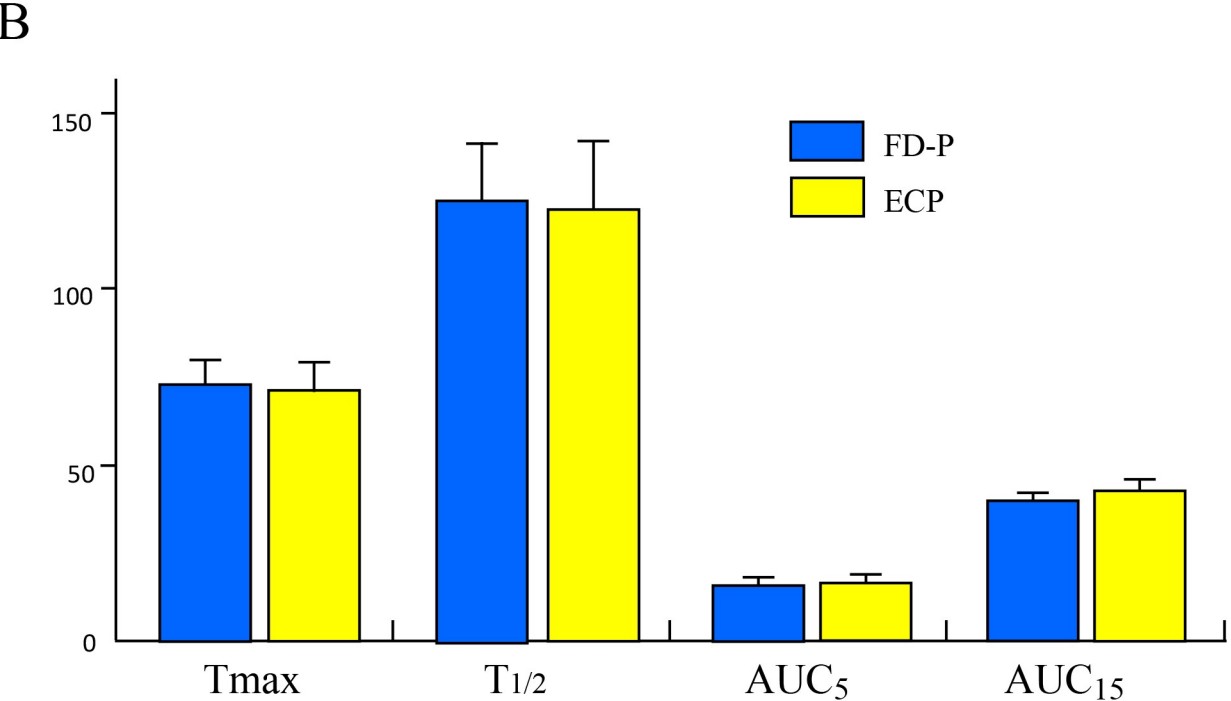

**Fig 1. Comparison of bothersome FD symptoms and clinical symptoms between ECP and FD-P patients. 1A.** There were not significant differences in epigastric pain, epigastric burning, postprandial fullness, early satiety, abdominal fullness, the feeling of hunger and discomfort for fat intakes between ECP and FD-P patients. **1B. Comparison of gastric motility between ECP and FD-P patients.** There were no significant differences in Tmax and T1/2 values between early chronic pancreatitis (ECP) and FD patients with pancreatic enzyme abnormalities (FD-P).

A

| Endosonographic features | FD-P (n=54) | ECP (n=26) | P value |
|---|---|---|---|
| Lobularity with honeycombing | 1/54 | 1/26 | <0.001 |
| Lobularity without honeycombing | 0/54 | 2/26 | 0.039 |
| Hyperechoic foci without shadowing | 3/54 | 13/26 | <0.001 |
| Stranding | 12/54 | 23/26 | <0.001 |
| Cysts | 1/54 | 0/26 | 0.485 |
| Dilated side branches | 0/54 | 3/26 | 0.011 |
| Hyperechoic main pancreatic ductal margin | 6/54 | 14/26 | <0.001 |

ECP: early chronic pancreatitis

FD-P: functional dyspepsia with pancreatic enzyme abnormalities

B

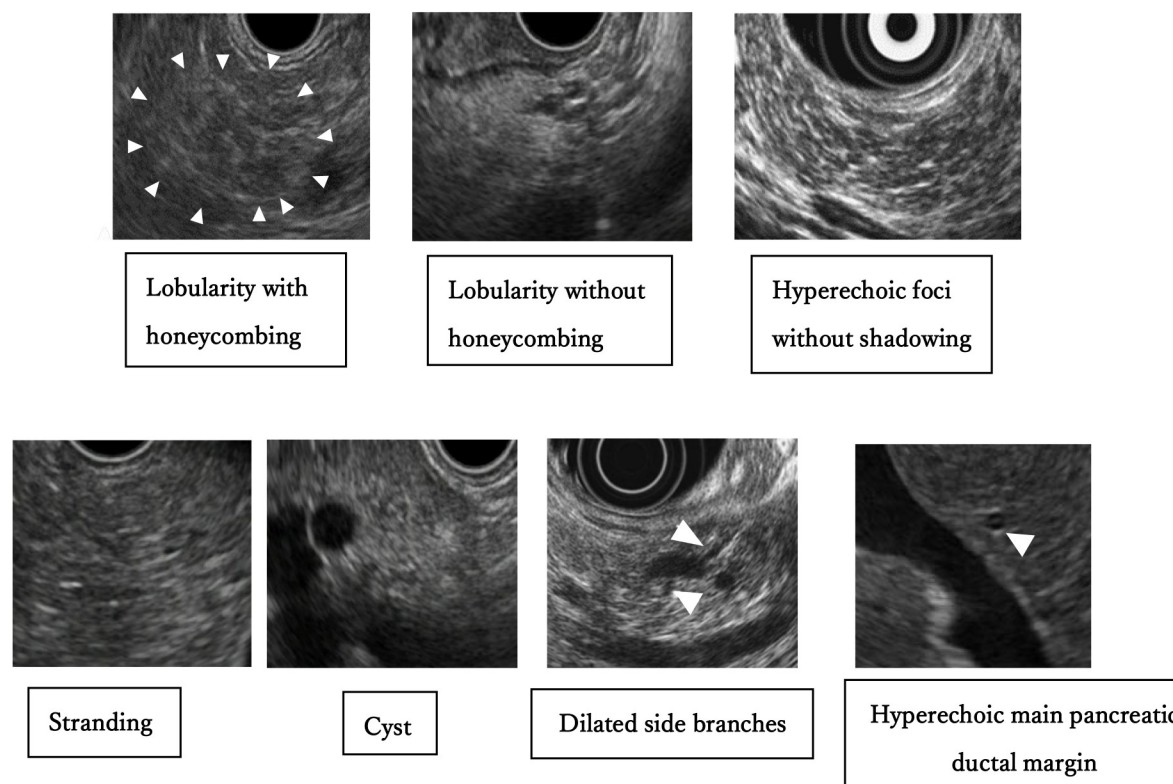

**Fig 2. Comparison of endosonographic features between ECP and FD-P patients.** Fig 2A We compared seven items of endosonographic features between FD-P and ECP patients. Fig 2B. We showed representative endosonographic features (lobularity with honeycombing, lobularity without honeycombing, hyperechoic foci without shadowing, stranding, cyst, dilated side branches, and hyperechoic main pancreatic ductal margin) of patients with ECP. Arrows show lobularity with honeycombing or dilated side branches, or hyperechoic main pancreatic ductal margin.

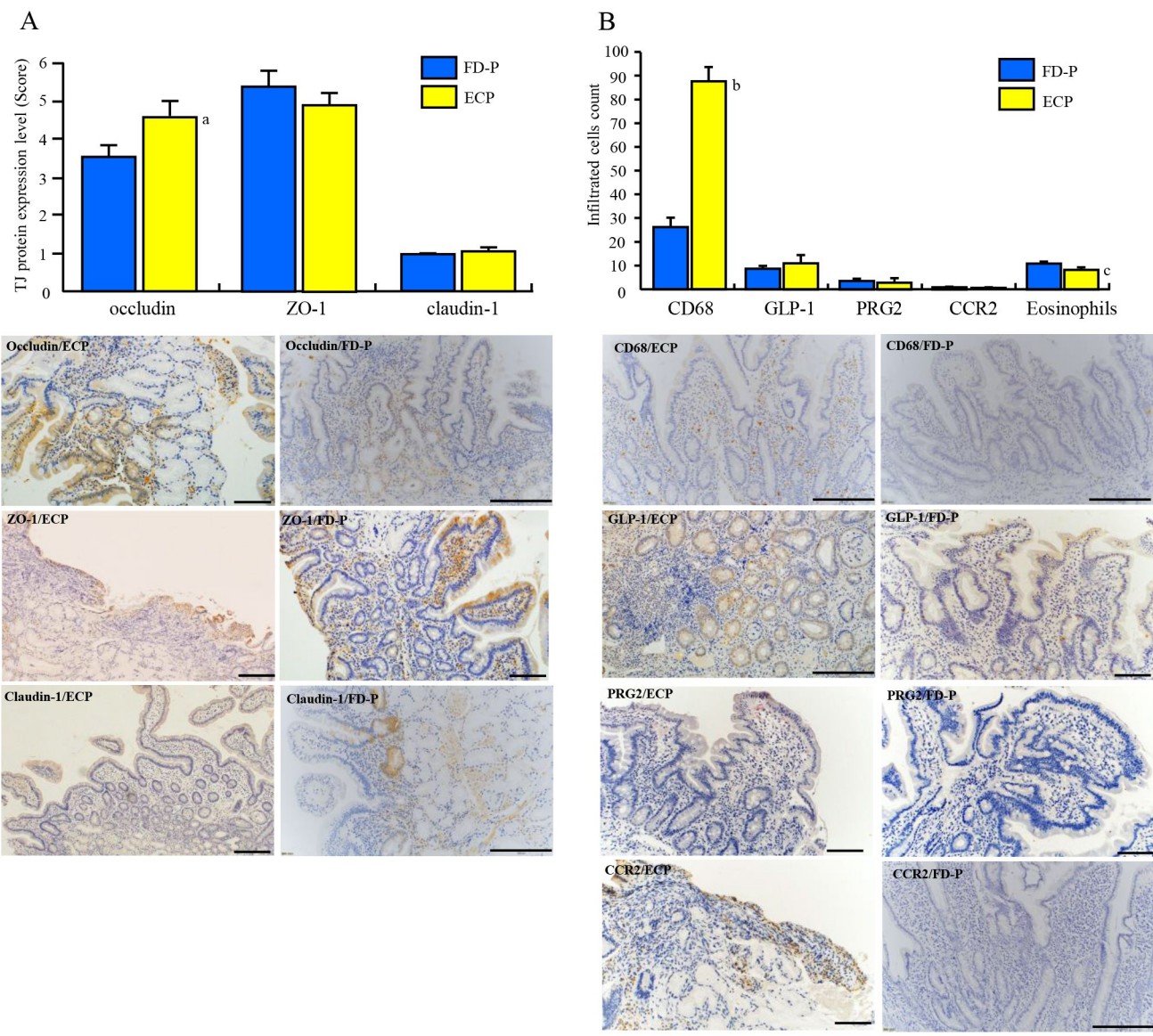

**Fig 3. Comparison of the expression levels of tight junction (TJ) proteins such as occludin, ZO-1 and claudin-1 between ECP and FD-P patients.**
**3A.** The expression levels of occludin were significantly higher in ECP patients than those patients with FD-P. In addition, there were not significant differences in ZO-1 and claudin-1 expression levels between ECP and FD-P patients. Representative occludin, ZO-1, and claudin-1 positive cells can be seen in patients with FD-P and ECP. a. vs FD-P patients, p = 0.048. **3B. Comparison of the expression levels of eosinophils, CD68-, GLP-1-, PRG2- and CCR2-positive cells between ECP and FD-P patients.** CD68-positive cells were significantly (p<0.01) higher in ECP patients than that of FD-P patients. Eosinophils were significantly (p = 0.008) higher in patients with FD-P than those of patients with ECP. However, there were not significant differences in GLP-1-positive cells, PRG2-positive cells and CCR2-positive cell between ECP and FD-P patients. Representative CD68-, GLP-1-, and CCR2-positive cells can be seen in patients with FD-P and ECP. b. vs FD-P patients, p<0.01. c. vs FD-P patients, p = 0.008. These groups were compared using two-tailed, unpaired, t-tests.

significant differences in ZO-1 (ECP:2±0.06, FD-P:2.05±0.08) and claudin-1(ECP:1.08±0.06, FD-P:1.00±0.01) expressions in the duodenal mucosa between ECP and FD-P patients (Fig 3A).

## Comparison of eosinophils, CD68-, GLP-1-, PRG-2-, and CCR2-positive cells infiltration in the duodenum between ECP and FD-P patients

Since duodenal inflammation was observed in patients with FD-P and ECP in our previous study, we compared the expression levels of inflammatory cells, including eosinophils, CD68-

positive cells, CCR2-positive cells, GLP-1 producing cells, and PRG-2 positive cells between the two groups. The CD68-positive cells counts (87.84±5.63) in the duodenum of ECP patients were significantly (p<0.01) higher than that of FD-P patients (26.78±3.96) (Fig 3B). In addition, eosinophils counts (11.19±0.81) in the duodenum of FD-P patients were significantly (p = 0.008) higher than those of ECP patients (8.81±0.75). However, there were no significant differences in the infiltrations of GL-P-1 positive cells (FD-P: 9.02±0.97, ECP: 11.39±3.4), degranulated eosinophils (FD-P: 3.55±0.74, ECP:3.19±1.48) and CCR-2-positive cells (FD-P:0.9 ±0.40, ECP:0.21±0.09) in the duodenum between patients with ECP and FD-P (Fig 3B).

## Correlation between CD68-possitive cells or degranulated eosinophils infiltration and the expression levels of occludin in ECP and FD-P patients

To determine whether duodenal inflammatory cells, such as CD68-positive cells or eosinophils, are associated with the expression levels of tight junction proteins, such as occludin, we compared their association in the duodenum of patients with ECP and FD-P. There was no significant correlation between CD68-positive cells infiltration and occludin expression levels in ECP ($R^2$ = 0.1, p = 0.66) and FD-P ($R^2$ = 0.1, p = 0.61) patients (Fig 4A and 4B). In addition, there was not significant correlation between degranulated eosinophils and occludin expression levels in ECP ($R^2$ = 0.31, p = 0.16) and FD-P patients ($R^2$ = -0.22, p = 0.27) (Fig 4C and 4D).

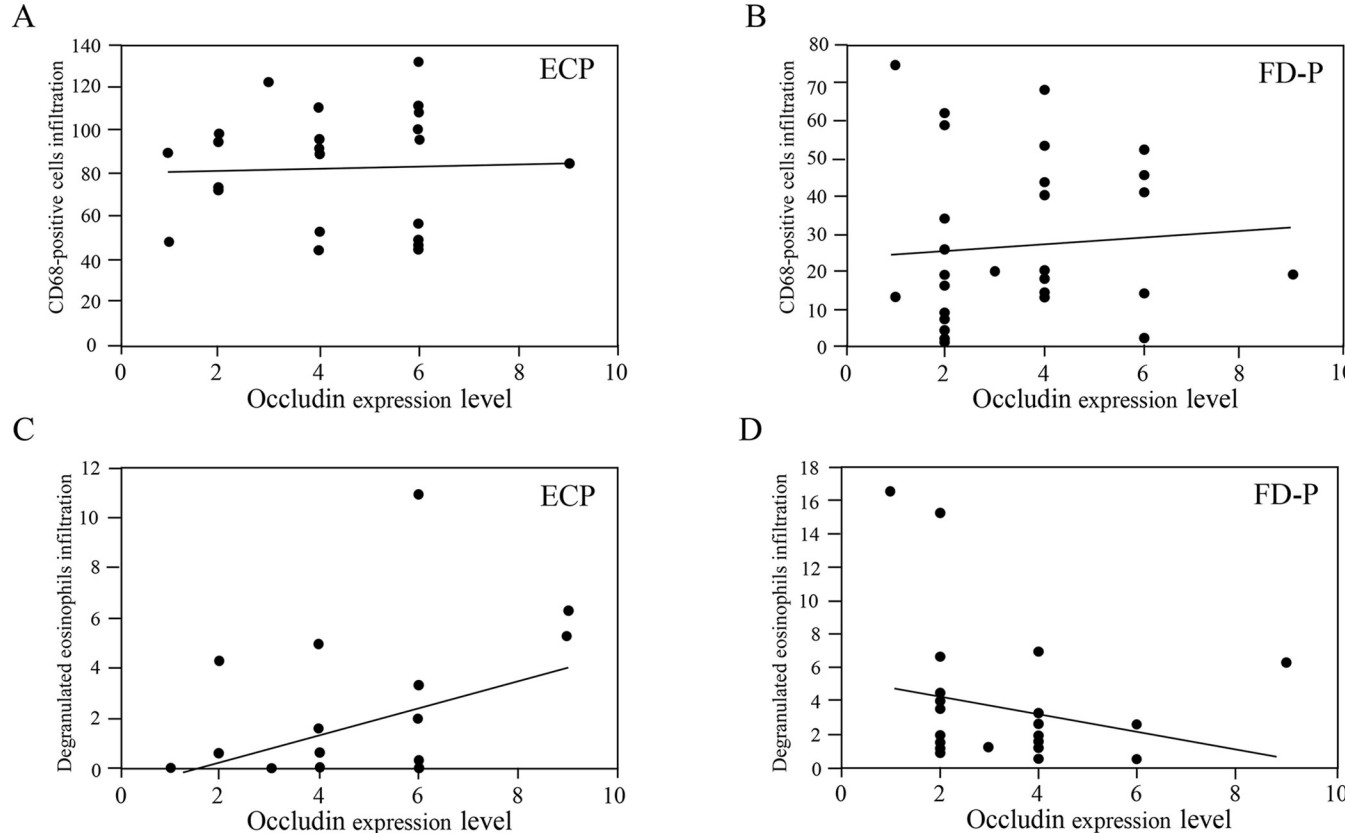

**Fig 4. Relationship between CD68-positive cells or degranulated eosinophils and occludin between ECP and FD-P patients.** There were not significant relationships between CD68-positive cells and occludin expression levels between ECP and FD-P patients (Fig 4A and 4B). In addition, there were not significant relationships between degranulated eosinophils and occludin expression levels between ECP and FD-P patients (Fig 4C and 4D). These groups were compared using a correlation coefficient.

## Trypsin levels were associated with peripheral and duodenal eosinophils in ECP and FD-P patients

To determine whether serum trypsin levels were associated with peripheral and duodenal eosinophils and duodenal inflammatory responses such as duodenal eosinophils count in ECP and FD-P patients, we compared serum trypsin levels with peripheral and duodenal eosinophils counts between ECP and FD-P patients. Interestingly, serum trypsin levels were significantly ($p = 0.007$) associated with peripheral eosinophils counts in patients with ECP (Fig 5A). In addition, serum trypsin levels were also significantly ($p < 0.05$ and $p < 0.001$, respectively) correlated with the duodenal eosinophils counts in ECP and FD-P patients (Fig 5A and 5B). Furthermore, 78.3% (20/26) of patients with ECP and 81.5% (44/54) of patients with FD-P exhibited elevated serum trypsin levels.

## Relationship between serum trypsin level and duodenal trypsin level in ECP and FD-P patients

Since positivity of abnormal trypsin level was highest among all pancreatic enzymes in ECP and FD-P patients, we focused on the relationship between serum and duodenal trypsin levels in ECP and FD-P patients. There was a mild relationship ($R^2 = 0.022$, $p = 0.36$) between serum trypsin levels and duodenal trypsin levels in patients with ECP and FD-P (Fig 6).

## Comparison of PAR2 mRNA levels in the duodenal mucosa of FD-P ECP patients

To determine the expression levels of the trypsin receptor, PAR2 in the duodenal mucosa of patients with ECP and FD-P, we compared PAR2 mRNA levels in duodenal tissues among ECP patients, FD-P patients and asymptomatic patients. PAR2 mRNA levels in duodenal mucosa of patients with ECP and FD-P were significantly ($p < 0.01$) increased compared to those in asymptomatic patients using real-time PCR (Fig 7). However, there were no significant differences in PAR2 mRNA levels in the duodenal mucosa between ECP and FD-P patients (Fig 7).

## Localization of PAR2 and degranulated eosinophils expression in the duodenum of patients with ECP and FD-P

To clarify the localization of PAR2 expression in the duodenal mucosa of patients with ECP and FD-P, we compared the localization of PAR2-positive cells in the duodenum between the two groups using immunostaining. PAR2 was expressed in the epithelial cells and the surface of mononuclear cells in the duodenal mucosa of ECP and FD-P patients (Fig 8A and 8B). In addition, degranulated eosinophils were localized beneath the epithelial cells (Fig 8C and 8D) in the duodenal mucosa of ECP and FD-P patients using anti-PRG2 antibody.

In Fig 9A, FITC-labeled (green) cells in the duodenum of patients with FD-P in show PAR2 immunoreactivity. Texas-red-conjugated cells in Fig 9A show EpCAM immunoreactivity for the same section (Fig 9A). There were many double-positive-stained cells (yellow cells) for the same section (Fig 9A). In addition, FITC-labeled (green) cells in the duodenum of patients with FD-P in Fig 9B show PRG2 immunoreactivity (Fig 9B). Texas-red-conjugated cells in Fig 9B show PAR2 immunoreactivity for the same section (Fig 9B). There were several double-positive-stained cells (yellow cells) for the same section (Fig 9B).

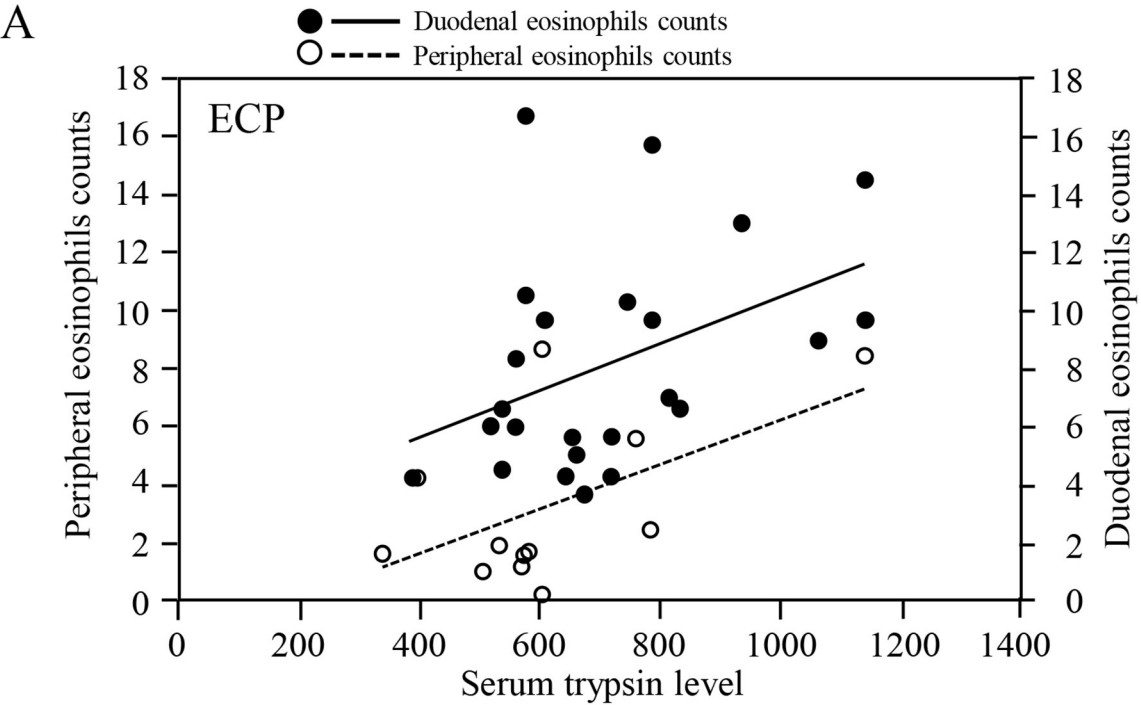

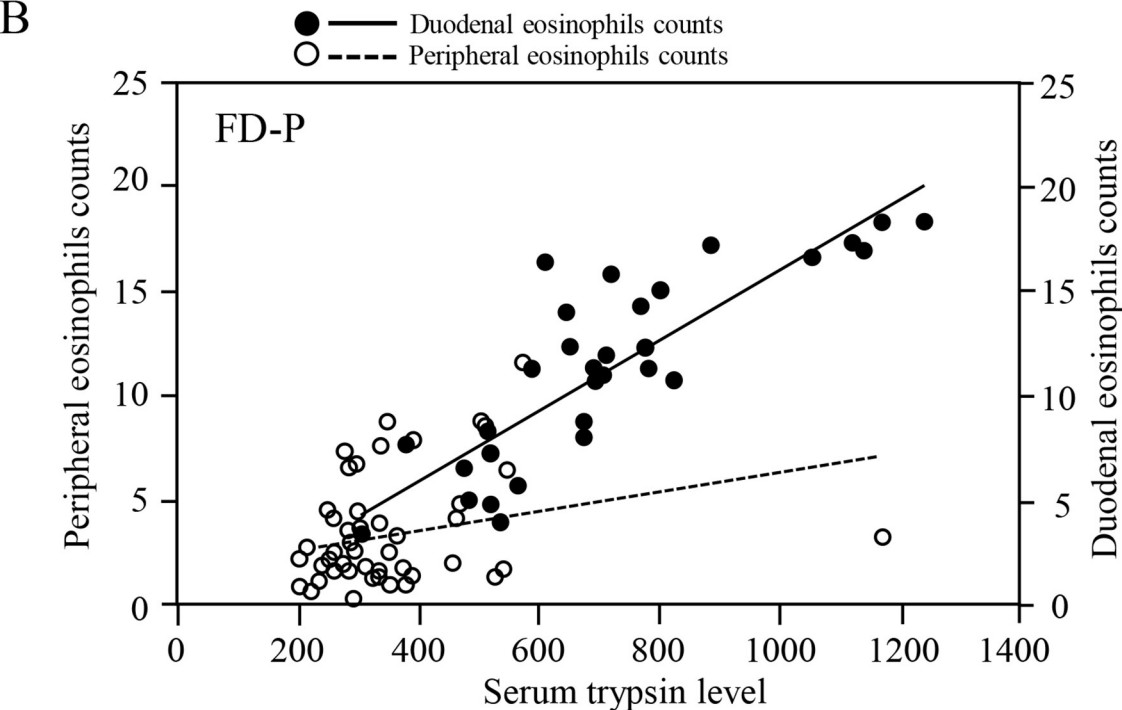

**Fig 5. Trypsin levels were associated with peripheral and duodenal eosinophils counts in ECP and FD-P patients.** Serum trypsin levels were significantly (p = 0.007, $R^2$ = 0.549) associated with peripheral eosinophils counts in patients with ECP (Fig 5A). In addition, serum trypsin level was not significantly (p = 0.05, $R^2$ = 0.267) associated with peripheral eosinophils counts in FD-P patients (Fig 5B). In addition, serum trypsin levels in ECP and FD-P patients were also significantly (p<0.05, $R^2$ = 0.419 and p<0.001, $R^2$ = 0.838, respectively) correlated with duodenal eosinophils counts (Fig 5A and 5B). These groups were compared using a correlation coefficient.

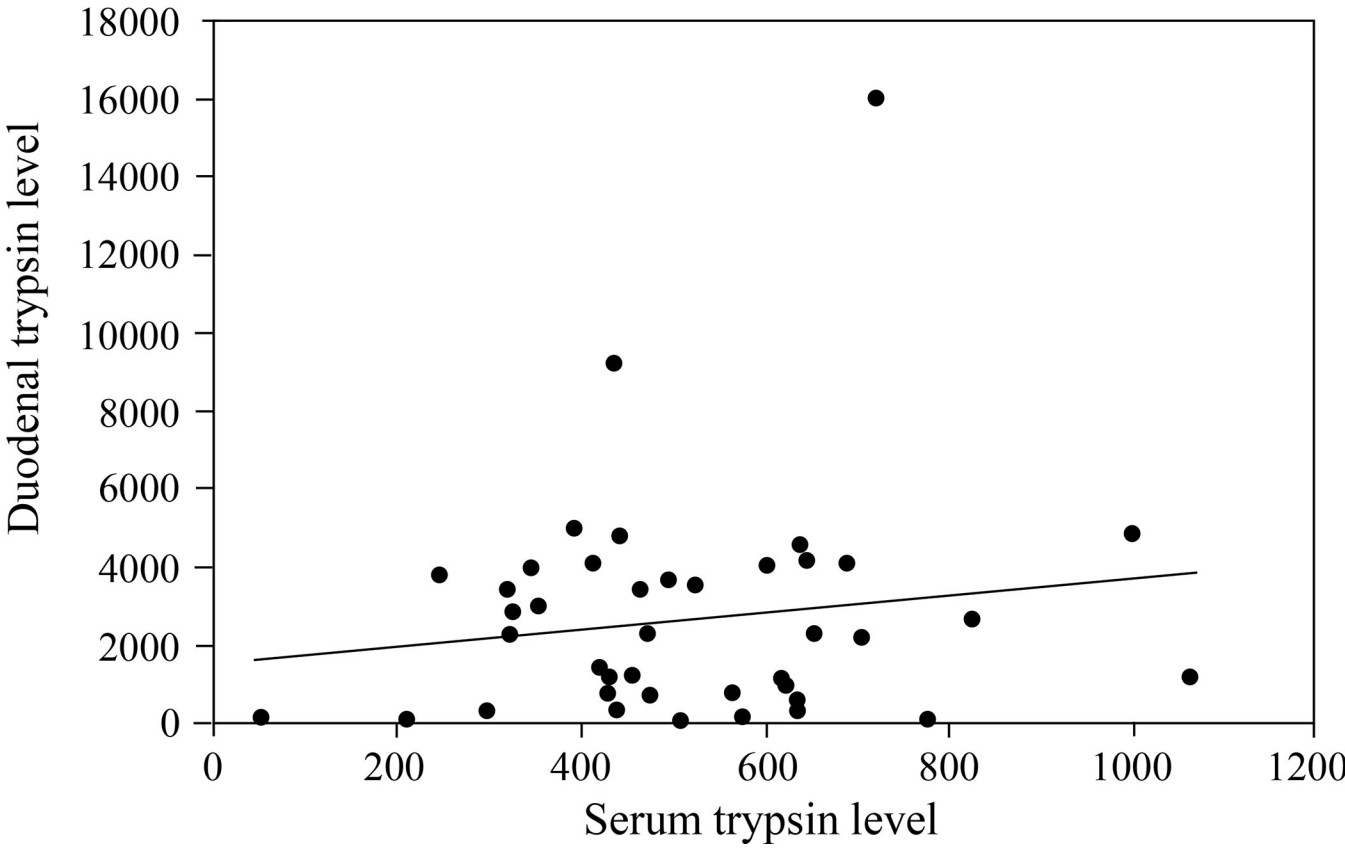

**Fig 6. Relationship between serum trypsin level and duodenal trypsin level in ECP and FD-P patients.** There was a mild relationship between serum trypsin levels and duodenal trypsin levels in patients with ECP and FD-P (Fig 6). These groups were compared using a correlation coefficient.

### Association between elevated trypsin levels and occludin or ZO-1 expression levels, and degranulated eosinophils infiltrations in the duodenum of patients with ECP and FD-P

To determine whether elevated trypsin levels (n = 64, 64/80) affected with the expression levels of tight junction proteins and the migration of degranulated eosinophils in patients with ECP and FD-P, we investigated whether elevated trypsin levels were associated with the expression levels of occludin and ZO-1 or with degranulated eosinophils in the duodenum of patients with ECP and FD-P. Elevated trypsin (64/80) were not significantly associated with occludin ($R^2$ = -0.27, p = 0.24 and $R^2$ = 0.06, p = 0.77), ZO-1 ($R^2$ = 0.23, p = 0.33 and $R^2$ = 0.31, p = 0.25) or degranulated eosinophils expression levels in the duodenum ($R^2$ = -0.21, p = 0.42 and $R^2$ = -0.14, p = 0.55) in patients with ECP and FD-P (S2A–S2F Fig).

## Discussion

Main findings in this study are 1) although there were no significant differences in clinical symptoms and gastric motility between ECP and FD-P patients, the scores of certain endoso-nographic features such as lobularity with honeycombing, hyperechoic foci without shadowing, stranding, dilated side branches, and hyperechoic main pancreatic ductal margin were significantly higher in ECP patients compared to those of FD-P patients, 2) CD68-positive cells infiltrations and the expression levels of occludin in the duodenum were significantly

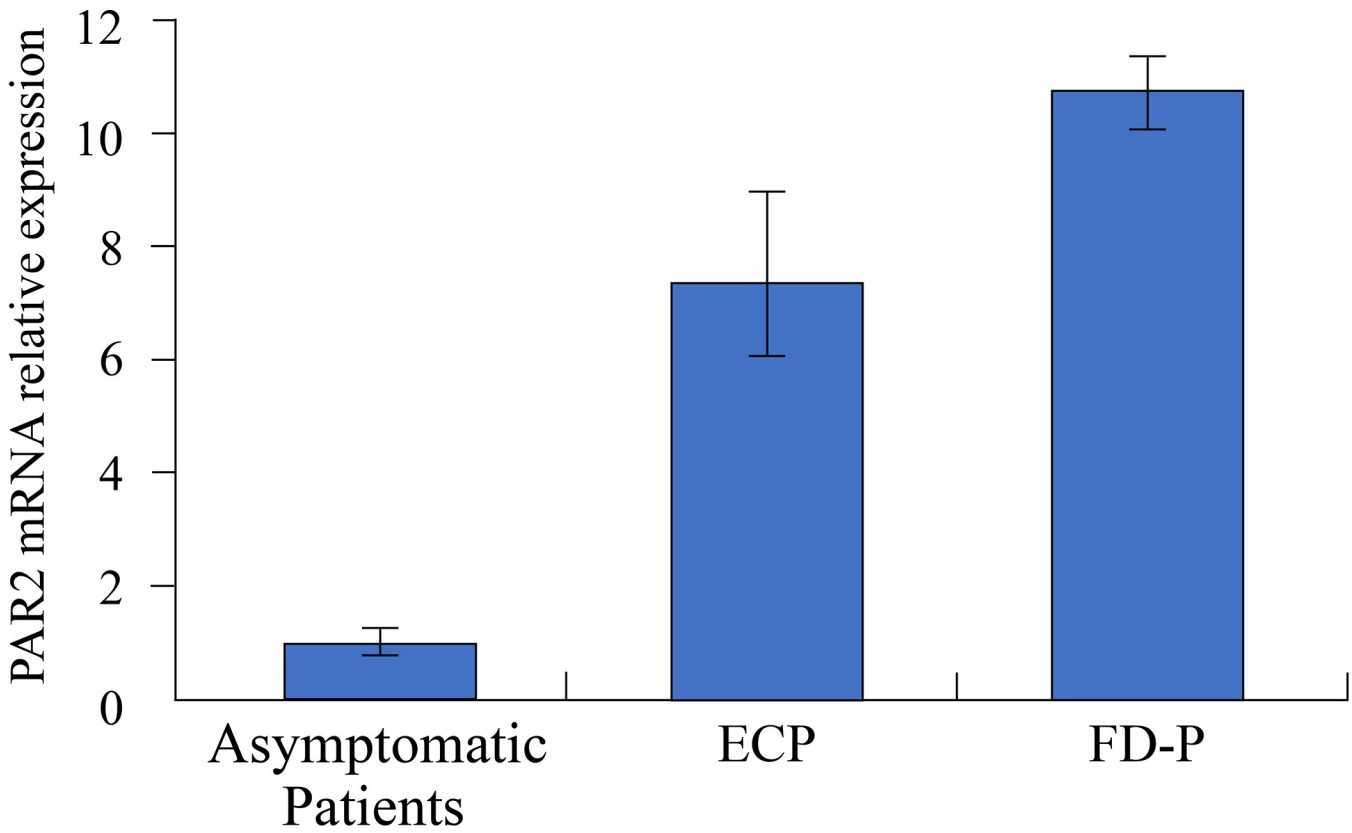

**Fig 7. Comparison of PAR2 mRNA levels in duodenal mucosa of FD-P patients with ECP patients.** PAR2 mRNA relative expression in the duodenal mucosa of patients with ECP (n = 4) and FD-P (n = 4) were significantly higher than those of asymptomatic patients (n = 12) using real-time PCR (Fig 7). However, there were no significant differences in PAR2 mRNA relative expression in the duodenal mucosa between ECP and FD-P patients (Fig 7). Results were compared using two-tailed, unpaired, t-tests.

reduced in FD-P patients compared to those of ECP patients, 3) serum trypsin levels in ECP patients were significantly (p = 0.007) associated with peripheral eosinophils and serum trypsin levels in ECP and FD-P patients were significantly (p<0.05 and p<0.001, respectively) associated with duodenal eosinophils respectively, 4) PAR2 expression localized in epithelial cells of the duodenum and degranulated eosinophils in patients with ECP and FD-P.

Since CP has a high mortality rate owing to a variety of factors, it is critical to identify CP at its early stage [28,29]. Some studies have reported that dyspeptic symptoms and symptoms consisting with irritable bowel syndrome may be the first and single clinical manifestation of chronic pancreatitis [30,31]. Therefore, it is very important to differentiate clinical characteristics and FD symptoms between these two cohorts of patients. Masamune et al have reported that the great majority of ECP cases did not develop into definite CP after two-year follow-up [32]. In addition, in the present study two cases (2/33) of patients with FD-P advanced into ECP patients, these may have actually been cases of ECP patients at an early stage. In our study, lobularity with honeycombing, hyperechoic foci without shadowing, stranding, dilated side branches, and hyperechoic main pancreatic ductal margin in endosonographic features were significantly higher in ECP patients compared to FD-P patients. We have also reported that non-alcoholic related ECP patients did not develop chronic pancreatitis. Diagnosis of FD-P patients based on no abnormal findings on upper GI endoscopy, abdominal US, and CT scan could not be completely excluded ECP patients without endosonography. In addition, in

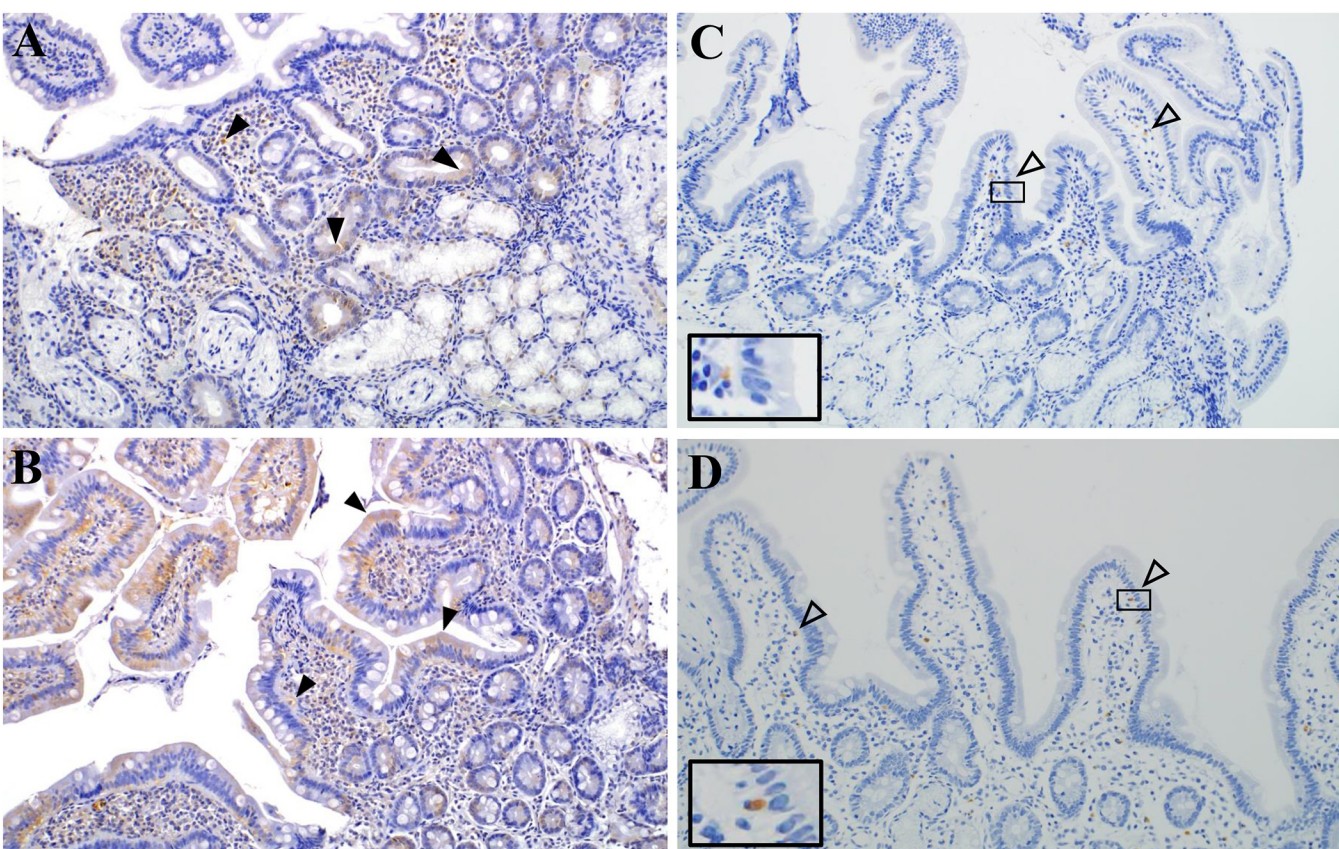

**Fig 8. Location of PAR2 expression in the duodenum of patients of FD-P and ECP.** PAR2 expression was localized in the duodenal epithelial cells and eosinophils in ECP and FD-P patients (Fig 8A and 8B). In addition, degranulated eosinophils were loacalized beneath the epithelial cells (Fig 8C and 8D) in the duodenal mucosa of patients with ECP and FD-P. Arrows (▼) showed PAR2-positive epithelial cells in the duodenum of patients with ECP and FD-P (Fig 8A and 8B, magnification: x200). Arrows (▽)demonstrated PRG2-positive degranulated eosinophils in the duodenum of patients with ECP and FD-P using anti-PRG2 antibody (Fig 8C and 8D, magnification: x200).

our data, exocrine pancreatic dysfunction in patients with FD-P was widely observed as well as previous study [33,34]. Exocrine pancreatic function is regulated by gut hormones such as CCK and 5-HT [35,36] and is associated with gastric emptying [37,38]. Therefore, FD-P concomitant with the impairment of gastric emptying induced by exocrine pancreatic dysfunction and abnormal activities of gut hormones, may be a prestage of early chronic pancreatitis. Further studies will be needed to clarify which types of FD-P patients will advance to ECP patients. Considering that the hyperechoic main pancreatic ductal margin, lobularity and dilated side branches vanished after treatment in our previous study [11], we hypothesize that certain endosonographic features described in Fig 1 may be reversible in ECP and FD-P patients.

The tight junction proteins are essential components of the intestinal barrier and consist of transmembrane proteins, such as occludin and claudin-1 [39]. Increased mucosal permeability, changes in tight junction proteins, and other pathological findings caused by slight inflammation have been studied in several gastrointestinal diseases [40–42]. Knockdown of occludin results in increased permeability, and the downregulation of occludin has been implicated in intestinal diseases [43]. Chang et al. have also reported that the expression of occludin and claudin-1 may be significantly reduced in FD rats by stress [44]. Lee et al. also reported that occludin expression levels were significantly lower in stressed rats than in controls [45]. Thus, the pathophysiology of patients with FD is reportedly associated with psychological stress.

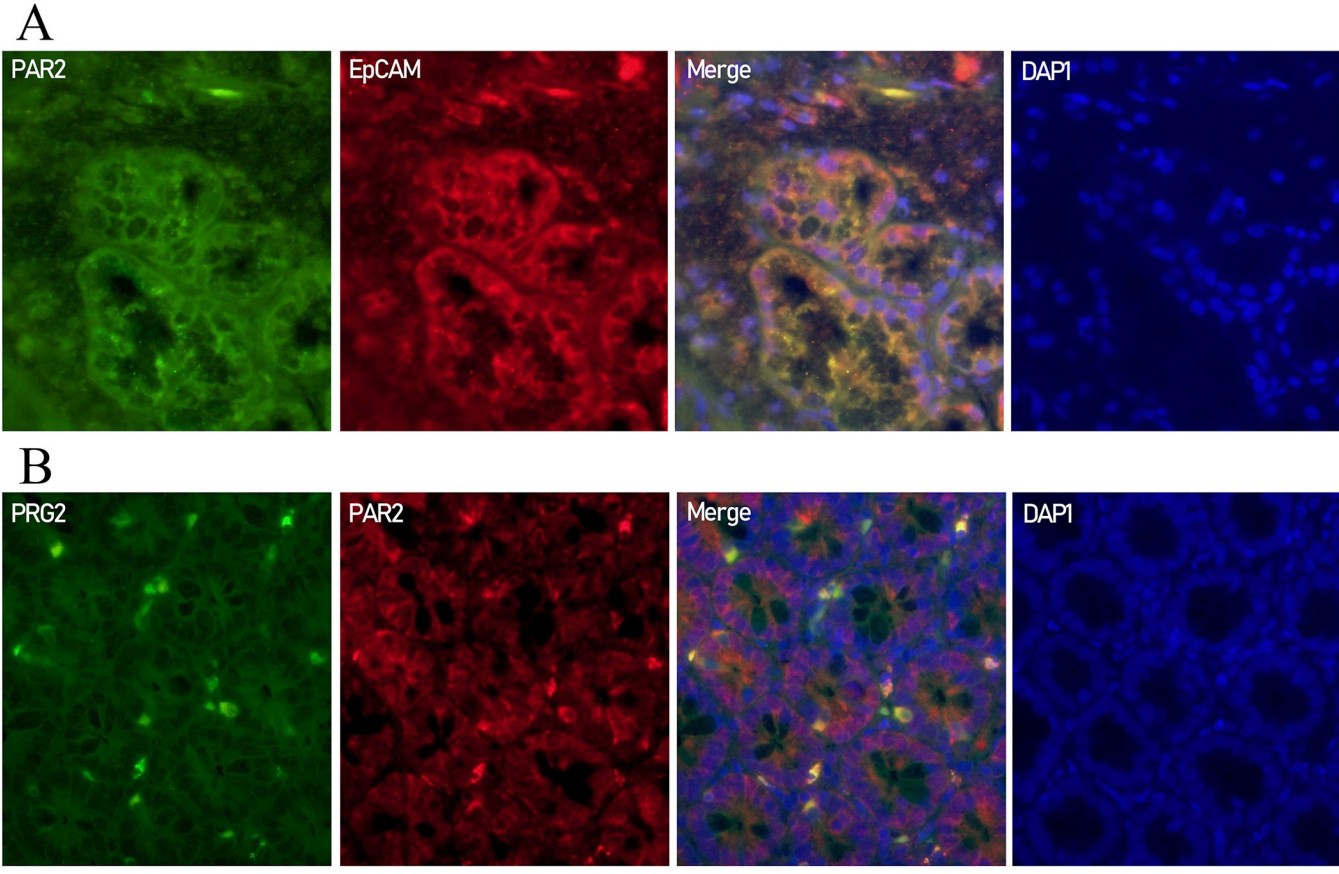

**Fig 9. Colocalization of PAR2 and EpCAM or PAR2 and degranulated eosinophils in the duodenum of patients with FD-P. Fig 9A.** PAR2-positive cells (green cells) in the duodenum of patients with FD-P. EpCAM-positive cells (red cells) in the duodenum of patients with FD-P. PAR2 and EpCAM expressions were mainly coexpressed in the same section. Double-positive cells (yellow cells) reveal PAR2- and EpCAM-positive cells in the same section. 4', 6-diamidino-2-phenylindole staining (blue cells). **Fig 9B.** PRG2-positive cells (green cells) in the duodenum of patients with FD-P. PAR2-positive cells (red cells) in the duodenum of patients with FD-P. PRG2 and PAR2 expressions were coexpressed in the same section. Double-positive cells (yellow cells) reveal PRG2- and PAR2-positive cells in the same section. 4', 6-diamidino-2-phenylindole staining (blue cells).

Considering of previous reports and our data, a significant reduction of occludin levels in the duodenum of patients with FD-P may also be associated with the stress. Cheng et al. have reported that claudin-1 expression levels were reduced in patients with diarrhea-predominant IBS [46]. Although ZO-1 and claudin expression levels in the duodenum have been reported to be associated with FD patients, respectively [47,48], in our data, there were no significant differences in ZO-1 and claudin-1 expression levels between FD-P and ECP patients.

Duodenal inflammation has also been linked to FD [12,49,50]. Vanheel et al discovered that not only eosinophils, but also mast cells, were found to be increased in the duodenum of patients with FD [49]. In our data, a significant elevation of eosinophils counts in patients with FD-P as described in Fig 3B may be partly reflected in the enhancement of duodenal permeability. However, there was no significant difference in the number of degranulated eosinophils between FD-P and ECP patients. These discrepancies between eosinophils and degranulated eosinophils counts in the duodenum may be dependent on the possibilities that degranulated eosinophils have been reported to be associated with the combination of eotaxin-3 and IL-5 [51]. Further studies will be needed to clarify the reason why duodenal inflammation occurred and was sustained in patients with FD. In addition, although we tried to clarify whether CD68-positive cells or granulated eosinophils were associated with the

reduction of tight junction proteins such as claudin-1, ZO-1, and occludin in the duodenum, CD68-positive cells and granulated eosinophils were not associated with tight junctions in the duodenum of patients with FD-P.

Reed et al. also reported that PAR2 stimulation of epithelial cells opens tight junctions, causes desquamation, and produces cytokines and growth factors [52]. Miike et al. reported that human eosinophils expressed mRNA for PAR2 and trypsin, an agonist for PAR2, induced degranulation of eosinophils [53]. Edogawa et al. also reported that the activation of PAR2 diminished the expression levels of tight junction proteins in patients with irritable bowel syndrome [54]. Considering previous studies and the significant relationship between trypsin levels and duodenal eosinophils, trypsin may be associated with eosinophils through the activation of PAR2 in the duodenal mucosa or on the surface of the eosinophils itself (graphical summary). In addition, trypsin levels may be associated with the reduction of occludin expression levels in the duodenum of patients with ECP through the activation of PAR2, albeit not statistically significant. Nestor et al. also reported that PAR2 expression has been reported in the colon of patients with IBS [55]. We are first to report that PAR2 mRNA and protein levels were determined in the duodenal mucosa of patients with FD-P and ECP (graphical summary). The association between duodenal degranulated eosinophils counts and trypsin levels did not reflect on a significant relationship between trypsin levels and peripheral eosinophils counts. Considering the above issues, eosinophil degranulation in the duodenum may require not only PAR2 stimulation with trypsin but also the existence of other inflammatory factors in the duodenum. The migration of duodenal degranulated eosinophils may require the coordinated activation of PAR2 and other inflammatory pathways, such as activated mast cells [56] or IL-33 production [57].

Taken together, our data suggest that elevated trypsin levels might be partly associated with duodenal inflammatory responses through PAR2-related duodenal epithelium and eosinophils infiltrations and the reduction of occludin in patients with ECP and FD-P.

## Supporting information

**S1 Checklist.**
(DOCX)

**S1 Fig. Study protocol.**
(TIF)

**S2 Fig. Relationship between elevated trypsin levels and occludin, ZO-1 and degranulated eosinophils expression levels between ECP and FD-P patients.** There was no significant relationship between elevated trypsin levels and occlusion or ZO-1 expression levels in ECP and FD-P patients (S2A, S2B, S2C, and S2D Fig). In addition, elevated trypsin levels were not associated with degranulated eosinophils infiltration in ECP and FD-P patients (S2E and S2F Fig). These groups were compared using a correlation coefficient.
(TIF)

## Author Contributions

**Data curation:** Shuhei Agawa, Seiji Futagami, Hiroshi Yamawaki, Rina Tsushima, Kazutoshi Higuchi, Mayu Habiro, Rie Kawawa, Yasuhiro Kodaka, Nobue Ueki, Yoshiyuki Watanabe, Katya Gudis, Rhuji Ohashi, Katsuhiko Iwakiri.

**Formal analysis:** Shuhei Agawa.

**Funding acquisition:** Seiji Futagami.

**Investigation:** Shuhei Agawa, Seiji Futagami.

**Writing – original draft:** Seiji Futagami.

**Writing – review & editing:** Seiji Futagami.

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
