## [Decision Letter · Decision Letter 0]

21 Apr 2022

PONE-D-21-36769Trypsin may be Associated with Duodenal Eosinophils through The Expression of PAR2 in Early Chronic Pancreatitis and FD with Pancreatic Enzyme AbnormalitiesPLOS ONE

Dear Dr. Futagami,

Thank you for submitting your manuscript to PLOS ONE. My sincere apologies for the delay in review. After careful consideration, we feel that it has merit but does not fully meet PLOS ONE’s publication criteria as it currently stands. Therefore, we invite you to submit a revised version of the manuscript that addresses the points raised during the review process.

The main concerns include the need for better imaging data, details of some of the protocols and patients as well as careful proofreading of the manuscript.  In addition authors should respond to all the individual comments of the reviewers.

We look forward to receiving your revised manuscript.

Kind regards,

Pradeep Dudeja

Academic Editor

PLOS ONE

Journal Requirements:

Reviewers' comments:

Reviewer's Responses to Questions

**Comments to the Author**

1. Is the manuscript technically sound, and do the data support the conclusions?

Reviewer #1: Partly

Reviewer #2: Yes

2. Has the statistical analysis been performed appropriately and rigorously? 

Reviewer #1: Yes

Reviewer #2: Yes

3. Have the authors made all data underlying the findings in their manuscript fully available?

Reviewer #1: Yes

Reviewer #2: Yes

4. Is the manuscript presented in an intelligible fashion and written in standard English?

Reviewer #1: Yes

Reviewer #2: Yes

5. Review Comments to the Author

Reviewer #1: Review comments :

1. Table 1 as mentioned in Results (Line 221) is not included in the manuscript.

2. The methodology for the determining the expression levels of tight junction proteins (Especially Line 187 – 188 is not clear).

3. Authors must include the histology staining images in Figure 2A and 2B in addition to the score graphs.

4. While there is no significant difference in the degranulated eosinophils between the two groups (FD-P and ECP), the eosinophil count in FD-P is significantly higher than in ECP. Does this indicate higher eosinophil degranulation rate in the ECP than in FD-P?

5. While serum trypsin levels were associated with peripheral and duodenal eosinophils (Figure 4), it is important to include details on what proportion of the patients (in each of ECP and FD-P group) had elevated or abnormal serum trypsin levels.

6. The authors must include the specific statistical used for determining significance for each Figure (in the figure legend or description).

7. Appropriate reference must be included for use of TGBC cell line as a control for ECP and FD-P duodenal tissue. The most ideal control would be healthy duodenal tissue of cadaveric origin. However, human primary duodenal cells available commercially could be more relevant than use of a Gall Bladder Cancer cell line (TGBC) used in this study.

8. The staining in Figure 7 does not say much about the cellular localization of PAR2. Co-staining of PAR2 with Epcam or other epithelial markers and with CD45 for mononuclear cells will add weightage to the claim.

9. Line No. 329 – 335 must be rephrased to make it understandable.

10. It is incorrectly mentioned in Line 345 (of Discussion) that trypsin levels in ECP and FD-P patients was significantly (p=0.007) associated with peripheral eosinophils. Significant association was found only in ECP group and not in FD-P group (Figure 4A).

11. While PAR2 is overexpressed in the duodenal mucosa of both ECP and FD-P patients and trypsin induced activation of PAR2 is attributed as the cause of reduced tight junction protein (Occludin) expression, why this effect (of reduced Occludin expression) is selectively observed in FD-P group and not in the ECP group is not explained.

12. Line 418 to 420 : Existence of other inflammatory pathways, involvement of mast cells and IL33 production is merely a speculation without any concrete proof or literature reference and its inclusion in the graphical abstract (mast cells) must be avoided.

13. H.pylori infection is one of the major cause of duodenal eosinophil degranulation and presence of this infection in the patients can majorly skew the observations and interpretation of the results. The authors must include the H.pylori infection status in the patient criteria.

14. While the authors mention that 80 patients were either classified as ECP or FD-P group, it would be interesting to include the information on total number of FD patients that were considered in this study, including those that did not fit in either of the two categories (Supplementary Figure 1).

15. The manuscript requires major rephrasing and rigorous spell check to make it lucid and avoiding confusion to the readers.

Reviewer #2: The manuscript by Agawa et al. makes a novel link between trypsin and early chronic pancreatitis in patients while the work describes original observations the following are suggested to substantiate the findings.

Major concerns

1. One of the major drawbacks of the study is lack of representative images to validate the main findings. For example, several endosonographic differences between FD-P and ECP were mentioned and none of the features were depicted as images for readers to visualize. In addition, this data is shown only in a supplementary figure and the data is described under a main heading in results section. Typically, main results should not go to supplementary materials

2. It is not convincing that the authors observed differences in tight junction proteins unless representative images are shown (Figure 2A). In addition, all the bar diagrams, it is advised that each data point be visible in the column graph since these are observations from clinical samples. The same is for CD 68+ cells

3. The PAR-2 staining images in Figure 7 should be co-stained with an eosinophilic marker to show co-staining

4. The graphical abstract can be improved and included as one of the main figures as well

Minor concerns

1. Please makes sure spelling mistakes are removed; E.g. Page 21 Lines 408 and 410 “Graphical” is incorrect. Please change to “graphical”

6. PLOS authors have the option to publish the peer review history of their article (what does this mean?). If published, this will include your full peer review and any attached files.

Reviewer #1: No

Reviewer #2: No

---

## [Author Response · Author response to Decision Letter 0]

27 May 2022

May 26th , 2022

PLoS One 

Editor-in-Chief,

Academic Editor, Prof. Pradeep Dudeja,

Dear Editor:

We would like to thank you for your letter of April 21th, 2022 permitting us to submit our revised manuscript (Manuscript No. PONE-D-21-36769), with the revised title of ” Trypsin may be Associated with Duodenal Eosinophils through The Expression of PAR2 in Early Chronic Pancreatitis and Functional Dyspepsia with Pancreatic Enzyme Abnormalities” by Agawa et al, for consideration for publication in PLoS One.

In response to the reviewers’ comments, we remade figures and Tables and added several changes to the manuscript. To respond to the reviewer’s comments properly, we added some statements in the revised version. Enclosed with this cover letter are our detailed answers to the reviewers in a point-by-point manner which include specific revisions. For convenience, all changes to the manuscript are noted by underline. 

We would like to take this opportunity to express our gratitude to the reviewers for their time and constructive and useful remarks. Their comments allowed us to identify areas in our manuscript that needed modification and clarification. We also take this opportunity to thank you personally for allowing us to submit a revised version of our manuscript. We hope that this revised manuscript is now acceptable for publication in PLoS One.

Sincerely yours,

Seiji Futagami, M.D., Ph.D. Prof.

Division of Gastroenterology, Nippon Medical School,

1-1-5, Sendagi, Bunkyo-ku, Tokyo, JAPAN

e-mail:seiji.futagami@gmail.com

Phone: +81-3822-2131　Fax: +81-5685-1793

 

Re:MS#PONE-D-21-36769,” Trypsin may be Associated with Duodenal Eosinophils through The Expression of PAR2 in Early Chronic Pancreatitis and Functional Dyspepsia with Pancreatic Enzyme Abnormalities” by Agawa et al.

Responses to the reviewer’s comments

Reviewer: 1

We thank the reviewer for the helpful and insightful suggestions in the review of our manuscript.

Reviewer #1: 

Major comments

Reviewer #1: Review comments :

1.Table 1 as mentioned in Results (Line 221) is not included in the manuscript.　

Based on the reviewer’s suggestions, we added Table 1 in the revised version. Please refer to it.

2.The methodology for the determining the expression levels of tight junction proteins (Especially Line 187 – 188 is not clear). 

According to the reviewer’s comments, we rewrote the sentences in the Methods section of the revised version (paragraph” Scoring of immunostaining for tight junction proteins in the duodenum”). Please refer to it.

3.Authors must include the histology staining images in Figure 2A and 2B in addition to the score graphs.

Based on the reviewer’s suggestions, we added the histology staining images in Figure 3A and Figure 3B in the revised version. Please refer to it.　

4.While there is no significant difference in the degranulated eosinophils between the two groups (FD-P and ECP), the eosinophil count in FD-P is significantly higher than in ECP. Does this indicate higher eosinophil degranulation rate in the ECP than in FD-P?

The permeability of duodenal mucosa of patients with FD have been reported to be increased in a previous study. We have also reported that the expression levels of occludin in patients with FD-P was significantly reduced than that in patients with ECP. Considering of our data, significant elevation of duodenal eosinophils counts in FD-P may be partly reflected on the enhancement of the permeability in the duodenum. However, there was no significant difference in the degranulated eosinophils between FD-P (26.3%) and ECP (21.3%) patients (p=0.42). Thus, eosinophil degranulation rate has been also reported to be associated with the cytokine levels such as eotaxin-1, eotaxin -2, and eotaxin-3 (Badewa AP, et al. Exp Biol Med, 2002, 227, 645-651) or the combination of eotaxin-3 and IL-5 (Badewa AP, et al. Immunopharmacology and Immunotoxicology, 25, 145-157, 2003). Considering of previous studies, these descrpancies between eosinophils counts and degranulated eosinophils counts in the duodenum may be partly depend on the possibilities that degranulated eosinophils have been reported to be associated with the combination of eotaxin-3 and IL-5. According to the reviewer’s suggestions, we added the above issues in the Discussion section of the revised version. Please refer to it.

5. While serum trypsin levels were associated with peripheral and duodenal eosinophils (Figure 4), it is important to include details on what proportion of the patients (in each of ECP and FD-P group) had elevated or abnormal serum trypsin levels. 

Based on the reviewer’s suggestions, we added the statements about the proportion of the patients had elevated serum trypsin levels (more than 550 ng/ml) in the Results section of the revised version (paragraph, “Trypsin levels were associated with peripheral and duodenal eosinophils in ECP and FD-P patients”). Please refer to it.

6. The authors must include the specific statistical used for determining significance for each Figure (in the figure legend or description).

Based on the reviewer’s suggestions, we added the statements about statistical analysis in the Methods section (paragraph, “Statistical Analysis”) and Figure legends in the revised version. Please refer to it. 

7. Appropriate reference must be included for use of TGBC cell line as a control for ECP and FD-P duodenal tissue. The most ideal control would be healthy duodenal tissue of cadaveric origin. However, human primary duodenal cells available commercially could be more relevant than use of a Gall Bladder Cancer cell line (TGBC) used in this study.

To respond the reviewer’s suggestions properly, we measured PAR2 mRNA levels in duodenal tissues from 12 asymptomatic patients without abnormal images and exchanged the data of TGBC51TKB cell-line into those from 12 asymptomatic patients. We remade revised Figure 7. Please refer to it. 

8. The staining in Figure 7 does not say much about the cellular localization of PAR2. Co-staining of PAR2 with Epcam or other epithelial markers and with CD45 for mononuclear cells will add weightage to the claim.

To respond to reviewer’s comments properly, we performed additional experiments. Using anti-Epcam antibody, we determined co-staining of PAR2 with Epcam in the duodenal specimens (Figure 9A). Moreover, using anti-PRG-2 antibody and anti-PAR2 antibody, we demonstrated co-staining of PAR-2 with degranulated eosinophils in the duodenum (Figure 9B). We added above sentences in the Results section of the revised version (paragraph, “Localization of PAR2 and degranulated eosinophils expression in the duodenum of patients with ECP and FD-P”). Please refer to it.

9. Line No. 329 – 335 must be rephrased to make it understandable.

To respond to reviewer’s comments properly, we rewrote the paragraph “Association between elevated trypsin levels and occludin or ZO-1 expression levels, and degranulated eosinophils infiltrations in the duodenum of patients with ECP and FD-P “in the Results section of the revised version. Please refer to it.

10. It is incorrectly mentioned in Line 345 (of Discussion) that trypsin levels in ECP and FD-P patients was significantly (p=0.007) associated with peripheral eosinophils. Significant association was found only in ECP group and not in FD-P group (Figure 4A).

We made a mistake for the statements about the relationship between trypsin levels in ECP and FD-P patients and peripheral eosinophils. We rewrote the relationship between trypsin levels in ECP patients and peripheral eosinophils. Please refer to it. 

11. While PAR2 is overexpressed in the duodenal mucosa of both ECP and FD-P patients and trypsin induced activation of PAR2 is attributed as the cause of reduced tight junction protein (Occludin) expression, why this effect (of reduced Occludin expression) is selectively observed in FD-P group and not in the ECP group is not explained.

Previous studies have also reported that occludin expression levels were significantly lower in stressed rat group compared to control group (Lee HS, et al. Gut and Liver, 7, 190-196, 2013) and Chang et al have also reported that occludin and claudin-1 expressions in the duodenum of FD rat might be reduced by stress (Chang X, et al. BMC Complementary and Alternative Medicine, 17, 432, 2017). Pathophysiology of patients with FD have been reported to be associated with psychological stress. Considering of previous studies and our data, occludin expression in the duodenum of patients with FD-P might be partly reduced by stress. According to the reviewer’s suggestions, we added above statements about the reduced occludin expression in the duodenum of patients with FD-P in the Discussion section of the revised version. Please refer to it.

12. Line 418 to 420 : Existence of other inflammatory pathways, involvement of mast cells and IL33 production is merely a speculation without any concrete proof or literature reference and its inclusion in the graphical abstract (mast cells) must be avoided.

Miwa et al have reported duodenal inflammation including eosinophils and mast cells plays pivotal roles of pathophysiology in patients with FD (Miwa H, et al J Gastroenterology, 54, 305-311, 2019). In addition, De Salvo et al have reported that IL-33 also induces eosinophil infiltrations in experimental ileitis. Based on the reviewer’s suggestions, we quoated above references in the statements in the Discussion section of the revised version. Please refer to it. In addition, to respond the reviewer’s suggestions properly, we cut the description including mast cells in the graphical abstract. Please refer to it. 

13. H.pylori infection is one of the major cause of duodenal eosinophil degranulation and presence of this infection in the patients can majorly skew the observations and interpretation of the results. The authors must include the H.pylori infection status in the patient criteria.

We agree with reviewer’s comments.　Based on the reviewer’s comments, we added the statements about H. pylori infection status in the Methods section (paragraph, “Patients”), Results section (paragraph, “Clinical characteristics of ECP and FD-P patients”), and new Table 1 in the revised version. We added the statement about H. pylori positivity in revised Table 1. Please refer to it. 

14. While the authors mention that 80 patients were either classified as ECP or FD-P group, it would be interesting to include the information on total number of FD patients that were considered in this study, including those that did not fit in either of the two categories (Supplementary Figure 1).

According to the reviewer’s comments, we remade revised supplementary Figure 1. In this study, we enrolled patients with 469 FD symptoms with no abnormal images. Then, we measured five kinds of pancreatic enzymes and eighty patients (80/469: 17.1%) with FD symtoms with pancreatic enzyme abnormalities without abnormal images were estimated using endosonography. Please refer to revised supplementary Figure 1.

15. The manuscript requires major rephrasing and rigorous spell check to make it lucid and avoiding confusion to the readers.

Based on the reviewer’s suggestions, a native speaker rewrote the revision to read easily.

Re:MS#PONE-D-21-36769,” Trypsin may be Associated with Duodenal Eosinophils through The Expression of PAR2 in Early Chronic Pancreatitis and Functional Dyspepsia with Pancreatic Enzyme Abnormalities” by Agawa et al.

Responses to the reviewer’s comments

Reviewer #2: The manuscript by Agawa et al. makes a novel link between trypsin and early chronic pancreatitis in patients while the work describes original observations the following are suggested to substantiate the findings.

Reviewer: 2

We thank the reviewer for the helpful and insightful suggestions in the review of our manuscript.

Major concerns

1. One of the major drawbacks of the study is lack of representative images to validate the main findings. For example, several endosonographic differences between FD-P and ECP were mentioned and none of the features were depicted as images for readers to visualize. In addition, this data is shown only in a supplementary figure and the data is described under a main heading in results section. Typically, main results should not go to supplementary materials.

To respond to the reviewer’s suggestions properly, we added representative images about endosonography of patients with ECP as revised Figure 2. We added the statements about representative images about endosonography in the Results section of the revised version (paragraph, “Comparison of endosonographic features between ECP and FD-P patients”). In addition, we also added the representative images about immunostaining in the revised Figure 3. Please refer to it. 

2. It is not convincing that the authors observed differences in tight junction proteins unless representative images are shown (Figure 2A). In addition, all the bar diagrams, it is advised that each data point be visible in the column graph since these are observations from clinical samples. The same is for CD 68+ cells

Based on the reviewer’s suggestions, we added the representative images about tight junction proteins and inflammatory cells such as CD68-positive cells in revised Figure 3. In addition, we described each data in the Results section of the revised version (paragraphs, “Comparison of tight junction proteins expression in the duodenal mucosa between FD-P and ECP patients” & “Comparison of eosinophils, CD68-, GLP-1, PRG-2-,and CCR2-positive cells infiltration in the duodenum between ECP and FD-P patients”). Please refer to it. 

3. The PAR-2 staining images in Figure 7 should be co-stained with an eosinophilic marker to show co-staining

To respond to reviewer’s comments properly, we performed additional experiments. Using anti-Epcam antibody, we determined co-staining of PAR2 with Epcam in the duodenal specimens (Figure 9). Moreover, using anti-PRG-2 antibody, we demonstrated co-staining of PAR2 with degranulated eosinophils in the duodenum of patients with FD-P (Figure 9). We added the statements about the colocalization of PAR2 and degranulated eosinophils in the Results section of the revised version (paragraph, “Localization of PAR2 and degranulated eosinophils expression in the duodenum of patients with ECP and FD-P”). Please refer to it.

4. The graphical abstract can be improved and included as one of the main figures as well

To respond to the reviewer’s suggestions properly, we remade the graphical abstract. Please refer to it.

Minor concerns

1. Please makes sure spelling mistakes are removed; E.g. Page 21 Lines 408 and 410 “Graphical” is incorrect. Please change to “graphical”

Based on the reviewer’s comments, we rewrote it. Please refer to it.

---

## [Decision Letter · Decision Letter 1]

20 Jul 2022

PONE-D-21-36769R1Trypsin may be Associated with Duodenal Eosinophils through The Expression of PAR2 in Early Chronic Pancreatitis and Functional Dyspepsia with Pancreatic Enzyme AbnormalitiesPLOS ONE

Dear Dr. Futagami,

Thank you for submitting your manuscript to PLOS ONE. After careful consideration, we feel that it has merit but does not fully meet PLOS ONE’s publication criteria as it currently stands. Therefore, we invite you to submit a revised version of the manuscript that addresses the points raised during the review process. While, one of the reviewer has no concerns, another reviewer has raised some minor points.  Addressing those issues should improve the overall manuscript and make it suitable for publication.

We look forward to receiving your revised manuscript.

Kind regards,

Pradeep Dudeja

Academic Editor

PLOS ONE

Journal Requirements:

Reviewers' comments:

Reviewer's Responses to Questions

**Comments to the Author**

1. If the authors have adequately addressed your comments raised in a previous round of review and you feel that this manuscript is now acceptable for publication, you may indicate that here to bypass the “Comments to the Author” section, enter your conflict of interest statement in the “Confidential to Editor” section, and submit your "Accept" recommendation.

Reviewer #2: All comments have been addressed

Reviewer #3: All comments have been addressed

2. Is the manuscript technically sound, and do the data support the conclusions?

Reviewer #2: Yes

Reviewer #3: Yes

3. Has the statistical analysis been performed appropriately and rigorously? 

Reviewer #2: Yes

Reviewer #3: Yes

4. Have the authors made all data underlying the findings in their manuscript fully available?

Reviewer #2: Yes

Reviewer #3: Yes

5. Is the manuscript presented in an intelligible fashion and written in standard English?

Reviewer #2: Yes

Reviewer #3: Yes

6. Review Comments to the Author

Reviewer #2: All concerns have been addressed. The manuscript can be now accepted for publication in the current form

Reviewer #3: Minor comments:

1. Table 1: it should be p-amylase

2. Figure 3a and b should have all staining figures for all genes and in both sets FD-P and ECP.

3. Statistical analysis line 223...please rephrase it

4. Figure 7 y axis needs revision...its relative gene mRNA level (normalized to internal control). please edit that

5. Please run spell check

7. PLOS authors have the option to publish the peer review history of their article (what does this mean?). If published, this will include your full peer review and any attached files.

Reviewer #2: No

Reviewer #3: No

---

## [Author Response · Author response to Decision Letter 1]

29 Aug 2022

August 27th , 2022

PLoS One 

Editor-in-Chief,

Academic Editor, Prof. Pradeep Dudeja,

Dear Editor:

We would like to thank you for your letter of July 21th, 2022 permitting us to submit our 2nd revised manuscript (Manuscript No. PONE-D-21-36769 R1), with the revised title of” Trypsin may be Associated with Duodenal Eosinophils through The Expression of PAR2 in Early Chronic Pancreatitis and Functional Dyspepsia with Pancreatic Enzyme Abnormalities” by Agawa et al, for consideration for publication in PLoS One.

In response to the reviewers’ comments, we remade figures and Tables and added several changes to the manuscript. To respond to the reviewer’s comments properly, we added some statements in the revised version. Enclosed with this cover letter are our detailed answers to the reviewers in a point-by-point manner which include specific revisions. For convenience, all changes to the manuscript are noted by underline. 

We would like to take this opportunity to express our gratitude to the reviewers for their time and constructive and useful remarks. Their comments allowed us to identify areas in our manuscript that needed modification and clarification. We also take this opportunity to thank you personally for allowing us to submit a revised version of our manuscript. We hope that this revised manuscript is now acceptable for publication in PLoS One.

Sincerely yours,

Seiji Futagami, M.D., Ph.D. Prof.

Division of Gastroenterology, Nippon Medical School,

1-1-5, Sendagi, Bunkyo-ku, Tokyo, JAPAN

e-mail:seiji.futagami@gmail.com

Phone: +81-3822-2131　Fax: +81-5685-1793

 

Re:MS#PONE-D-21-36769R1,” Trypsin may be Associated with Duodenal Eosinophils through The Expression of PAR2 in Early Chronic Pancreatitis and Functional Dyspepsia with Pancreatic Enzyme Abnormalities” by Agawa et al.

Responses to the reviewer’s comments

Reviewer: 3

We thank the reviewer for the helpful and insightful suggestions in the review of our manuscript.

Reviewer #3: 

Minor comments

Reviewer #3: Review comments:

1. Table 1: it should be p-amylase

Based on the reviewer’s comments, we rewrote p-amylase in the revised Table 1. Please refer to it.

2. Figure 3a and b should have all staining figures for all genes and in both sets FD-P and ECP.

According to the reviewer’s comments, we remade Figure 3a and 3b. We added all staining

figures in both sets of FD-P and ECP. Please refer to it.

3. Statistical analysis line 223...please rephrase it.

To respond to reviewer’s comments properly, we rewrote the paragraph “Statistical analysis”. Please refer to it.

4. Figure 7 y axis needs revision...its relative gene mRNA level (normalized to internal control). please edit that.

Based on the reviewer’s suggestions,　We rewrote y axis unit in Figure 7 to PAR2 mRNA relative expression. Please refer to it.

5. Please run spell check

Based on the reviewer’s suggestions, we rewrote, and a native speaker checked the revision. Please refer to it.

---

## [Editor Report · Decision Letter 2]

14 Sep 2022

Trypsin may be Associated with Duodenal Eosinophils through The Expression of PAR2 in Early Chronic Pancreatitis and Functional Dyspepsia with Pancreatic Enzyme Abnormalities

PONE-D-21-36769R2

Dear Dr. Futagami,

We’re pleased to inform you that your manuscript has been judged scientifically suitable for publication and will be formally accepted for publication once it meets all outstanding technical requirements.

Kind regards,

Pradeep Dudeja

Academic Editor

PLOS ONE
---

## [Editor Report · Acceptance letter]

6 Oct 2022

PONE-D-21-36769R2 

Trypsin may be Associated with Duodenal Eosinophils through The Expression of PAR2 in Early Chronic Pancreatitis and Functional Dyspepsia with Pancreatic Enzyme Abnormalities 

Dear Dr. Futagami:

I'm pleased to inform you that your manuscript has been deemed suitable for publication in PLOS ONE. Congratulations! Your manuscript is now with our production department. 

Kind regards, 

on behalf of

Dr. Pradeep Dudeja 

Academic Editor

PLOS ONE